# Cancer Cell-Derived Granulocyte-Macrophage Colony-Stimulating Factor Is Dispensable for the Progression of 4T1 Murine Breast Cancer

**DOI:** 10.3390/ijms20246342

**Published:** 2019-12-16

**Authors:** Teizo Yoshimura, Kaoru Nakamura, Chunning Li, Masayoshi Fujisawa, Tsuyoshi Shiina, Mayu Imamura, Tiantian Li, Naofumi Mukaida, Akihiro Matsukawa

**Affiliations:** 1Department of Pathology and Experimental Medicine, Graduate School of Medicine, Dentistry and Pharmaceutical Sciences, Okayama University, 2-5-1 Shikata, Kita-ku, Okayama 700-8558, Japan; pv0j6hsw@s.okayama-u.ac.jp (K.N.); 18640512230@163.com (C.L.); mfujisawa@okayama-u.ac.jp (M.F.); pp842ts2@s.okayama-u.ac.jp (T.S.); ptnm7qkv@s.okayama-u.ac.jp (M.I.); opheliali2012@163.com (T.L.); amatsu@md.okayama-u.ac.jp (A.M.); 2Division of Molecular Bioregulation, Cancer Research Institute, Kanazawa University, Kakuma-machi, Kanazawa 920-1192, Japan; mukaida@staff.kanazawa-u.ac.jp

**Keywords:** macrophages, chemokines, cytokines, inflammation, tumor microenvironment, breast cancer

## Abstract

We previously reported that 4T1 murine breast cancer cells produce GM-CSF that up-regulates macrophage expression of several cancer promoting genes, including *Mcp-1/Ccl2*, *Ccl17* and *Rankl*, suggesting a critical role of cancer cell-derived GM-CSF in cancer progression. Here, we attempted to define whether 4T1 cell-derived GM-CSF contributes to the expression of these genes by 4T1tumors, and their subsequent progression. Intraperitoneal injection of anti-GM-CSF neutralizing antibody did not decrease the expression of *Mcp-1*, *Ccl17* or *Rankl* mRNA by 4T1 tumors. To further examine the role of cancer cell-derived GM-CSF, we generated GM-CSF-deficient 4T1 cells by using the Crisper-Cas9 system. As previously demonstrated, 4T1 cells are a mixture of cells and cloning of cells by itself significantly reduced tumor growth and lung metastasis. By contrast, GM-CSF-deficiency did not affect tumor growth, lung metastasis or the expression of these chemokine and cytokine genes in tumor tissues. By in-situ hybridization, the expression of *Mcp-1* mRNA was detected in both *F4/80*-expressing and non-expressing cells in tumors of GM-CSF-deficient cells. These results indicate that cancer cell-derived GM-CSF is dispensable for the tuning of the 4T1 tumor microenvironment and the production of MCP-1, CCL17 or RANKL in the 4T1 tumor microenvironment is likely regulated by redundant mechanisms.

## 1. Introduction

Tumor microenvironments consist of not only tumor cells but also non-tumor stromal cells, including fibroblasts, endothelial cells, myocytes and inflammatory cells, such as macrophages, regulatory T cells (Tregs), dendritic cells (DCs) and myeloid-derived suppressor cells (MDSCs). The crosstalk between tumor cells and stromal cells leads to the production of an array of mediators that provide the soil for tumor cells to grow, invade and metastasize [1,2,3]. Therefore, identifying the mechanisms by which tumor cells and non-tumor cells crosstalk in tumor microenvironments is critical for the treatment of cancer patients.

The chemokine monocyte chemoattractant protein-1 (MCP-1)/CCL2 is a chemoattractant that plays an important role in the recruitment of blood monocytes into tumors and known to promote lung metastasis of breast cancer cells [4,5]. In human breast cancer patients and the murine 4T1 breast cancer model, non-tumor stromal cells were the main source of MCP-1 that promotes lung metastasis [5,6], and granulocyte-macrophage colonystimulating factor (GM-CSF/CSF-2) produced and released by 4T1 cells highly induced MCP-1 production by mouse inflammatory macrophages in vitro [7]. In addition to MCP-1, GM-CSF selectively upregulated the expression of other genes by macrophages whose products play an important role in the development of breast cancer, including CCL17 and tumor necrosis factor superfamily member 11 (TNFSF11; also known as RANKL) by recruiting Tregs [8] and facilitating mammary tumorigenesis [9], respectively. These previous results led us to hypothesize that GM-CSF produced by breast cancer cells may contribute to the establishment of tumor microenvironments by inducing the production of cancer-promoting cytokines/chemokines, such as MCP-1, CCL17 and RANKL, by tumor-infiltrating macrophages.

In the present study, we examined the contribution of 4T1 cell-derived GM-CSF to the tuning of the tumor microenvironment by examining the expression of the *Mcp-1*, *Ccl17* and *Rankl* gene by 4T1 tumors. For the purpose, we first neutralized GM-CSF activity in tumor-bearing mice. We next generated GM-CSF-deleted 4T1 cells by using the CRIPR-Cas9 system [10]. Although GM-CSF-neutralization or GM-CSF-deletion in cancer cells did not reduce the expression of these genes or the progression of tumors, cloning of single cells from the parental cells dramatically affected the expression of those three genes or the progression of 4T1 tumors. Thus, we provide novel information about the role of cancer cell-derived GM-CSF and a potential cooperation among heterogenous cancer cells in the progression of cancer.

## 2. Results

### 2.1. Neutralization of GM-CSF Does not Affect the Expression of Mcp-1, Ccl17 or Rankl mRNA by 4T1 tumors

To determine whether 4T1 cell-derived GM-CSF contributes to the expression of *Mcp-1*, *Ccl17* or *Rankl* mRNA by 4T1 tumors, we neutralized GM-CSF by intraperitoneal injection of anti-GM-CSF antibody and examined the expression of these genes by 4T1 tumors two weeks after the transplantation of tumor cells. We chose two weeks because the serum MCP-1 concentration reached a peak in tumor-bearing mice at 2 weeks and the necrosis of tumor tissues, that could affect the expression of these genes, became more evident after this time. As shown in Figure 1, neutralization of GM-CSF did not significantly affect the weight of the primary tumors or the expression of the *Mcp-1*, *Ccl17* or *Rankl* genes by the tumors.

### 2.2. Generation of GM-CSF-Deficient 4T1 Cells Using the CRISPR-Cas9 System

To obtain additional information as to the role of cancer cell-derived GM-CSF in the progression of the 4T1 breast cancer, we generated GM-CSF-deficient 4T1 cells by using the CRISPR-Cas9 system. We transfected 4T1 cells with either control or GM-CSF double nickase plasmid. After the selection of transfected cells in the presence of puromycin and single cell cloning, we obtained multiple clones without GM-CSF production as determined by ELISA. Some of the clones were further analyzed to confirm the presence of indels in the targeted region of the *Gm-csf* gene (Figure 2).

We selected two control clones (C1 and C2) and 4 GM-CSF-deficient clones (KO1-4) for further studies (Figure 3). Morphologically, the parental 4T1 cells consisted of a mixture of epithelial-like cells attaching firm to the plastic surface and round shape cells weakly attaching to the surface (Figure 3A). By contrast, each clone we obtained was uniform but exhibited slightly different cell shapes. C1 and C2 cells attached to the plastic surface firmly and were both epithelial-like. KO1, 2 and 4 cells were also epithelial-like, whereas KO3 cells remained round after plating and weakly attached to the plastic surface (Figure 3A). C1 cells produced a slightly higher level of GM-CSF compared to that of the parental 4T1 cells, whereas C2 cells produced a much higher level of GM-CSF (Figure 3B). GM-CSF was not detected in the culture supernatants of all four KO clones by ELISA (Figure 3B) and the capacity of their supernatants to induce MCP-1 production in macrophages was greatly reduced compared with those of the parental or control clone cells (Figure 3C). 

### 2.3. Single Cell Cloning of 4T1 Cells Reduced the Progression of 4T1 Breast Cancer, Independent of GM-CSF Production

We first examined the growth of each clone in vitro and found that all clones, except for KO3, grew significantly slower than the parental cells (Figure 4A). To evaluate the in vivo growth and metastasis of these 4T1 clones, we injected 1 × 10^5^ cells of each clone into the mammary pad of female BALB/c mice and the tumor weight, the number of lung metastases and the weight of spleens were evaluated 4 weeks after the injection. As shown in Figure 4B, the growth of all six individual clones tended to be slower than that of the parental 4T1 cells, and the weight of C1, KO1, KO3 and KO4 tumor was significantly lower. The number of lung metastases by all six clones was significantly reduced compared with that of the parental cells. The weight of spleens was lower in mice bearing C1, KO1 and KO3 tumors. Thus, although there were some differences among clones, there was no clear correlation between the level of GM-CSF production and the growth, lung metastasis and spleen weight. These results indicated that the progression of 4T1 tumor was not dependent on cancer cell-derived GM-CSF. 

### 2.4. The Expression of Mcp-1/Ccl2, Ccl17 or Rankl by 4T1 Tumors Is Independent of Tumor Cell-Derived GM-CSF 

We next evaluated the expression of *Mcp-1*, *Ccl17* and *Rankl* mRNA by each tumor 2 weeks after the transplantation of tumor cells. As shown in Figure 5A,B, there was no significant difference in the level of *Mcp-1* mRNA or serum MCP-1 concentration among tumor groups. The expression of *Ccl17* mRNA was increased in C1 tumors and the expression of *Rankl* mRNA was increased in C2 tumors. However, again there was no correlation between the level of *Ccl17* or *Rankl* mRNA and the level of GM-CSF produced by each clone in vitro (Figure 5B). There was no detectable level of GM-CSF by ELISA in the sera from mice bearing tumors of the parental 4T1 or the clones used in this study. 

We also evaluated the infiltration of Ly6G-positive neutrophils, F4/80-positive macrophages and FoxP3-positive Tregs in tumors of the original 4T1, C1, C2, KO1 and KO2 cells by immunohistochemistry (IHC). Transplantation of each cell resulted in the formation of solid tumors and the infiltration of leukocytes was detected in all tumors by H&E-staining (Figure 6A). By immunohistochemistry, similar numbers of these leukocyte subsets were detected in all tumors and there was no correlation between the level of GM-CSF production in vitro and the infiltration of these cell types in the tumors (Figure 6B).

### 2.5. Tumor-Infiltrating Macrophages Express MCP-1 mRNA Independent of Cancer Cell-Derived GM-CSF

Our results presented above strongly suggested that cancer cell-derived GM-CSF is dispensable for the progression of 4T1 tumors and the tuning of the tumor microenvironment. It was also possible that tumor-infiltrating macrophages express and produce MCP-1 independently of cancer cell-derived GM-CSF. Therefore, we attempted to determine whether MCP-1 was produced by macrophages infiltrating the KO4 tumors. We first used IHC to detect MCP-1 protein but our attempts with multiple antibodies against mouse MCP-1 were unsuccessful. This could be due to rapid secretion of MCP-1 from cells. We therefore used in situ hybridization (ISH) to detect *Mcp-1* mRNA. A large number of F4/80-positive macrophages were detected between cancer cells by immunohistochemistry (Figure 7A). By ISH, the expression of *Adgre1 (F4/80)* mRNA (shown as red dots) was detected between cancer cells and some of them were also positive for *Mcp-1* mRNA (shown as green dots) (Figure 7B, indicated by arrows). This result indicated that tumor infiltrating macrophages express *Mcp-1* mRNA independent of cancer cell-derived GM-CSF.

### 2.6. The Level of Cancer Cell-Derived G-CSF Correlates with Splenomegaly and Tissue Congestion 

As described above, the level of GM-CSF production did not correlate with the degree of splenomegaly. To examine the role of G-CSF in this process, we used 4 individual clones from the parental 4T1 cells among which D5 cells expressed a significantly higher level of *G-csf* mRNA in vitro (Figure 8A). Transplantation of D5 caused a significantly higher level of splenomegaly (Figure 8B) and tissue congestion (Figure 8C), suggesting that the overproduction of G-CSF, but not GM-CSF, contributes to splenomegaly and tissue congestion.

## 3. Discussion

GM-CSF was first defined as a factor with its ability to generate colonies of mature granulocytes and macrophages from bone marrow precursor cells, but it also acts on mature cells [11]. Increased production of GM-CSF is implicated in the pathogenesis of inflammatory or autoimmune diseases and inhibiting this cytokine or its receptor appears to be a beneficial therapy for patients with the diseases [11]. GM-CSF has also been used as an adjuvant to stimulate immune responses against cancer [12]. There have been several reports suggesting that it promotes breast cancer metastasis to remote organs. Takeda et al. demonstrated a correlation between *Gm-csf* gene expression and metastasis by using 14 transplantable murine tumors, including breast cancers [13]. Park et al. examined the role of NF-kB in osteolytic bone metastasis of the human MDA-MB-231 breast cancer cells and found that GM-CSF secreted by breast cancer cells promoted osteolytic bone metastasis [14]. Recently, mesenchymal-like human breast cancer cells were demonstrated to produce GM-CSF and cancer cell-derived GM-CSF transformed macrophages to a TAM-like phenotype and to produce the chemokine CCL18, promoting the lung and liver metastasis in humanized mice [15]. We previously reported that GM-CSF activates macrophages in vitro to express a unique set of cytokine and chemokine genes, including *Mcp-1*, *Ccl17* and *Rankl* genes, whose products are demonstrated to promote the progression of breast cancer [7]. GM-CSF induces MCP-1 expression via Jak-Stat5 signaling [16]. In the present study, we evaluated the role of cancer cell-derived GM-CSF in the progression of breast cancer by first neutralizing its activity in vivo and by injecting them into the mammary pad of female mice. Neutralization of GM-CSF did not significantly change tumor growth, lung metastasis or spleen weight. There was also no correlation between the level of GM-CSF production by each cancer cell clones and the expression of *Mcp-1*, *Ccl17* and *Rankl* gene or the progression of tumors. Thus, our results indicate that cancer cell-derived GM-CSF is dispensable for the progression of the 4T1 breast cancer.

Since GM-CSF induced a high level of MCP-1 production in inflammatory macrophages in vitro, we hypothesized that the production of MCP-1 in the 4T1 tumor microenvironment is dependent on cancer cell-derived GM-CSF. However, our results indicate that macrophages infiltrating 4T1 tumors produce MCP-1 by mechanisms independent of GM-CSF. In fact, by ISH *Mcp-1* mRNA was detected in both F4/80-positive and negative cells in tumors of GM-CSF-deficient 4T1 clone. It is well known that activation by other ligands also leads to the production of MCP-1 by macrophages. These ligands include TLR ligands, IL-1 and TNFα [17]. TNFα is a tumor-promoting cytokine [18] and TNF-deficiency in stromal cells significantly decreased MCP-1 production by tumors of the mouse Lewis lung cancer cells [18]. Unlike GM-CSF, TNFα can activate many types of cells, including hematopoietic and non-hematopoietic cells, to produce MCP-1. Therefore, although the level of the *Tnfα*mRNA expression in 4T1 tumors may not be as high as in LLC tumors as previously reported [19], TNFα could still be a better target to reduce the production of MCP-1 and other cancer promoting cytokines. Additional studies are necessary to determine the role of TNFα in this breast cancer model.

A leukemoid reaction with granulocytosis and splenomegaly has been observed in animal cancer models and human patients with a variety of tumors. 4T1 tumor-bearing mice exhibit a leukemoid reaction, and it may be caused by the production of colony-stimulating factors, such as GM-CSF and G-CSF, by the tumor [20]. It was previously demonstrated that the inoculation of leukemic 12B1 cells transduced to produce a high level of GM-CSF causeed splenomegaly and extensive foci of congested lungs [21], suggesting a role of GM-CSF in this process. In our study, a similar degree of splenomegaly was detected in mice bearing either GM-CSF-producing or deficient 4T1 clones. Furthermore, we previously reported that neutralization of GM-CSF did not reduce the weight of spleens [7]. Thus, GM-CSF does not play a significant role in splenomegaly in this mode. On the other hand, transplantation of a 4T1-derived clone with high *G-csf* mRNA expression caused a markedly higher level of splenomegaly and tissue congestion. Recently, G-CSF-deficient 4T1 cells were generated and inoculated into mice. Similar to the GM-deficiency, the G-CSF deficiency did not affect tumor establishment or growth in vivo; however, there was a marked reduction in the sizes and weights of spleens, indicating a role of 4T1 cell-derived G-CSF in the development of splenomegaly in this breast cancer model [22]. 

It was recently demonstrated that 4T1 is not a single clone cells but a mixture of many types of cells with different abilities to grow and metastasize [23]. The original parent tumor of 4T1 was isolated from a single spontaneously arising mammary tumor of a MMTV^+^ BALB/c mouse foster nursed on a C3H mother (BALB/BfC3H) [24]. The line 410 was obtained from a metastatic nodule isolated from a BALB/cfC3H mouse bearing the tenth subcutaneous passage of the original parent tumor. 4T1 is a subline derived from the 410.4 tumor, that was derived from the fourth transplant generation of the line 410 [25,26,27]. Thus, although the line 4T1 was established after a few selection steps described above, this line comprises of heterogenous clones and a cooperation among the clones, rather than the production of single cytokine, such as GM-CSF, appears to influence the tumor progression. 

In conclusion, tumor tissues consist of many types of cells and the production of tumor promoting cytokines is regulated by many redundant mechanisms. However, the production of these cytokines could be regulated differently in each tumor microenvironment and GM-CSF may play more significant role in other cancer models. As described above, GM-CSF plays important roles in many human diseases and animal disease models, including cancer, and the balance between its pro- and anti-tumor activities may determine the final outcome. Additional studies using this and also other cancer models will provide important information to determine strategies to treat patients with cancer. 

## 4. Materials and Methods 

### 4.1. Reagents

RPMI-1640 was from Sigma-Aldrich (St. Louis, MO, USA) and Nakarai Tesque (Kyoto, Japan). Phosphate buffered saline (PBS), citric acid and 0.5M EDTA solution were from Wako Pure Chemical Corp. (Osaka, Japan). TRIsure™ reagent was from Nippon Genetics (Tokyo, Japan). Fetal bovine serum (FBS) was from HyClone (Logan, UT, USA). Normal rat IgG was from MP Biomedicals (Santa Ana, CA, USA) and neutralizing antibody against mouse GM-CSF (clone MP122E9) was from BioLegend (San Diego, CA, USA). Thioglycollate (TG) was from Difco Laboratories (Detroit, MI, USA). L-Glutamine, penicillin/streptomycin, trypsin-EDTA, sodium pyruvate solution and RNAlater were from Life Technologies (Gaithersburg, MD, USA). Proteinase K and the High Pure RNA Isolation Kit were from Roche (Mannheim, Germany). Anti-Ly6G rat monoclonal IgG (clone 1A8) and anti-FoxP3 rat monoclonal IgG (clone MF-14) were from BioLegend. Anti-F4/80 rat monoclonal IgG (clone BM8) was from eBioscience (San Diego, CA, USA).

### 4.2. Transplantation of 4T1 Cells and Neutralization of GM-CSF

Wild type female BALB/c mice were purchased from Japan SLC, Inc. (Hamamatsu, Japan). 4T1 cells (ATCC, Manassas, VA, USA) were cultured in RPMI 1640 supplemented by 10% FBS, 2 mM L-glutamine, penicillin/streptomycin and sodium pyruvate. One million cells were grown to 50 to 80% confluence in T-75 tissue culture flasks. Cells were detached with 0.2% trypsin-EDTA, washed once with medium, three times with PBS and resuspended in PBS at 1 × 10^6^/mL. One hundred μL of cell suspension (1 × 10^5^ cells) were injected into the 3rd mammary pad of female mice. Mice were separated into 2 groups; 5 mice in the control group received i.p. injection of normal rat IgG (100 μg in 100 μL PBS) whereas 5 mice in the experiment group received anti-mouse GM-CSF IgG (100 μg in 100 μL PBS) twice a week, for 2 weeks. Blood was collected by heart puncture, and sera were isolated and stored at −80 °C until use. Tumors were harvested and a half was fixed in 10% neutral buffered formalin (Wako) and the other half was in RNAlater. Lungs were perfused with Bouin’s solution (Wako), fixed in the same solution, and then the number of tumor nodules was counted by eye. Tumor length and width were measured using a caliper and tumor volume was calculated using the following formula: Volume = (width)^2^ × length/2. 

### 4.3. Generation and Transplantation of GM-CSF-Deficient 4T1 Cell Lines 

To stably delete the expression of GM-CSF in 4T1 cells, 1 × 10^4^ cells were seeded into a 6-well plate. After overnight incubation at 37 °C, cells were transfected with GM-CSF Double Nickase Plasmid or Control Double Nickase Plasmid (Santa Cruz Biotechnology, Inc.) [10] using Lipofectamine 3000 (Invitrogen, Carlsbad, CA, USA). Sequences for sgRNAs used to disrupt the *Gm-csf* gene were as follows; 5′-CAAAGAAGCCCTGAACCTCC-3′ and 5′-CAAGGCCGGGTGACAGTGAT-3′.

Seventy-two hours after transfection, medium was changed to the same medium containing puromycin (2 μg/mL) and transfected cells were selected for 5 days in the presence of puromycin. Subsequently single cell clones were selected through serial dilution. The presence of indels in each clone was verified by PCR, followed by DNA sequencing.

For in vivo transplantation studies, cells were prepared, injected and tumors were harvested as described above. To generate culture supernatants of parental 4T1 and 4T1 clones, 1 × 10^5^ cells in 2 mL RPMI1640 containing 10% FBS were seeded in 24-well plates and incubated at 37 °C for 3 days and cell-free supernatants were obtained by centrifugation and stored at −80 °C until use.

### 4.4. Activation of Macrophages in Vitro

Peritoneal exudate cells (PECs) were harvested by peritoneal lavage using 5 mL cold PBS 4 days after the intraperitoneal injection of 1 mL 3% TG. After centrifugation, PECs were resuspended in RPMI 1640 containing 10% FBS, penicillin and streptomycin, at the concentration of 2.5 × 10^6^, and then cultured for indicated times in the presence or absence of 50% (*v*/*v*) culture supernatants.

### 4.5. Quantification of Cytokine/Chemokine Concentration

The concentrations of GM-CSF or MCP-1 were measured using ELISA kits specific for mouse GM-CSF or MCP-1 (BioLegend). 

### 4.6. qRT-PCR

The expression of cytokine and chemokine genes was assessed by qRT-PCR. Total RNA was extracted from RNAlater-fixed tumor tissues and non-activated or activated PEC by using TRIsure or High Pure RNA Isolation Kit. The quality and yield were assessed by Nanodrop spectrophotometry, and then cDNA was synthesized using High Capacity cDNA Reverse Transcription Kit (Applied Biosystems, Foster City, CA, USA). qRT-PCR was performed using the StepOne Plus Real-Time PCR system (Life Technologies). The expression of the *Mcp-1, Ccl17, Rankl* and *Gapdh* gene was analyzed by Taqman gene expression assays (Applied Biosystems). The expression level of each gene was normalized to that of the *Gapdh* gene and presented as fold change over the expression of control gene.

### 4.7. Immunohistochemistry (IHC)

Tumors were fixed overnight with 10% formalin and embedded in paraffin. Serial sections of 4 μm thickness were prepared from formalin-fixed and paraffin-embedded blocks collected as described above. One section was stained with H&E and others were used for immunohistochemistry for each block. Immunostaining was performed manually by a conventional method: Briefly, sections were deparaffinized in xylene and rehydrated in a sequence of descending concentrations of ethanol. Endogenous peroxidase reactivity was blocked with 3% H_2_O_2_ for 10 min. For antigen retrieval, sections were submerged in 0.1 M citrate buffer (pH6.0) or 5 mM EDTA solution (pH8.0) and microwaved (500W) continuously for 20 min in a pressure cooker. For F4/80 staining, sections were treated with 200 μg/mL proteinase K (Roche) for 5 min at room temperature. Sections were then incubated with a respective primary antibody for 1.5 h at room temperature. After washing, sections were treated with Polink-2 Plus HRP RAT or RABBIT with DAB kit (GBI, Inc., Bothell, WA, USA) according to the manufacturer’s instructions. Finally, sections were counterstained with hematoxylin, dehydrated, and mounted. Images were acquired using an Olympus BX43 light microscope connected to a DP73 digital camera (Olympus, Tokyo, Japan). 

### 4.8. In Situ Hybridization (ISH)

Expression of the *Mcp-1/Ccl2* or *adhesion G protein-coupled receptor E1* (*Adgre1*, also known as the gene coding for F4/80) mRNA in tumor tissues was detected by RNAScope® 2.5 Duplex Detection Kit (ACD, Inc., Hayward, CA, USA). Briefly, tissue sections in 5-μm thickness were deparaffinized in xylene, followed by dehydration in an ethanol series. Tissue sections were then incubated in citrate buffer (10 nmol/L, pH 6) maintained at a boiling temperature (100 °C to 103 °C) using a hot plate for 15 min, rinsed in deionized water, and immediately treated with protease at 40 °C for 30 min in a Dako StatSpin Hybridizer (Agilent, Santa Clara, CA, USA). The tissue sections were then incubated at 40 °C in a hybridization buffer containing target probes (mouse Ccl2, C1 probe, #311791 and mouse Adgre1, C2 probe, #460651-C2) for 4 h. After the hybridization step, slides were washed with wash buffer three times at room temperature, and then label probe of each amplification system were used. Chromogenic detection was performed using FastRed or FastGreen, followed by counterstaining with hematoxylin. Images were acquired using an Olympus BX43 light microscope connected to a DP73 digital camera. 

### 4.9. Statistical Analysis

Statistical analysis was performed by Student’s *t*-test using the GraphPad Prism, Version 4 and 5 (GraphPad Software, San Diego, CA, USA). A value of *p < 0.05* was considered to be statistically significant.

### 4.10. Ethics Statement

The experimental protocols of this study were approved (OKU-2015548, 2019033) by the Animal Care and Use Committee of the Okayama University, Okayama, Japan, and all experiments were performed in accordance with relevant guidelines and regulations.

## Figures and Tables

**Figure 1 ijms-20-06342-f001:**
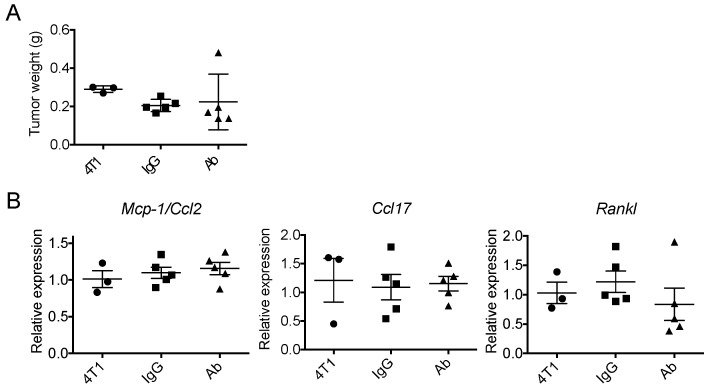
Anti-GM-CSF treatment of 4T1 tumor-bearing mice did not affect tumor weight or the expression of the *Mcp-1/Ccl2, Ccl17 or Rankl* genes. One hundred μg of either control rat IgG or anti-GM-CSF Ab was intraperitoneally injected on day 0, 3, 7 and 10. Mice were euthanized on day 14. (**A**) Tumors were harvested from the mice and the weight of each tumor was weighed. (**B**) The expression of *Mcp-1/Ccl2, Ccl17* and *Rankle* mRNA was evaluated by qRT-PCR. The results are shown as the mean ± SEM. *n* = 3 for untreated group. *n* = 5 for IgG- or Ab-treated group.

**Figure 2 ijms-20-06342-f002:**
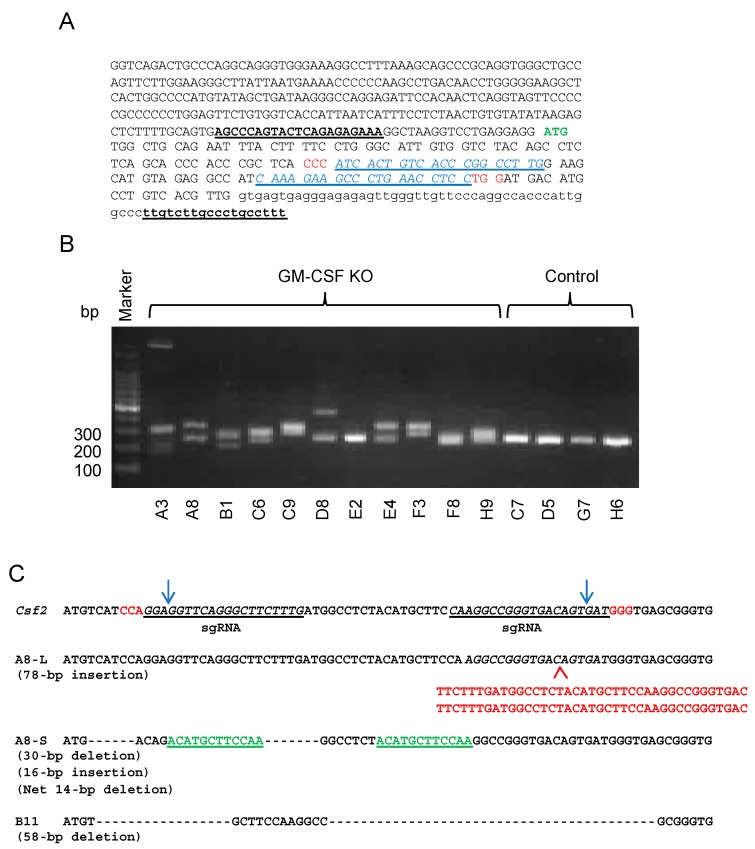
Generation of GM-CSF-deficient 4T1 cells. (**A**) The genomic sequence of the targeted region. ATG in green indicates the initiation codon. The PAM sequences are indicated in red. The sequences in blue indicate the sequences for guide RNA. The sequences with underline indicate the primers for PCR. (**B**) Genomic DNA from each clone was subjected to PCR to detect the presence of indels. (**C**) The presence of indels was confirmed by DNA sequencing of the PCR products. The presented sequence is the reverse complement of the coding sequence shown in (**A**). The PAM sequences are indicated in red. Deletions (---) and insertions (in red below the A8-L sequence) were detected within the targeted region. A8-S had a 16-bp insertion containing a part of the original sequence indicated in green. Clone A8 is KO1 in this study. Clone B11 is KO2 in this study but not included in the photo presented as (**B**).

**Figure 3 ijms-20-06342-f003:**
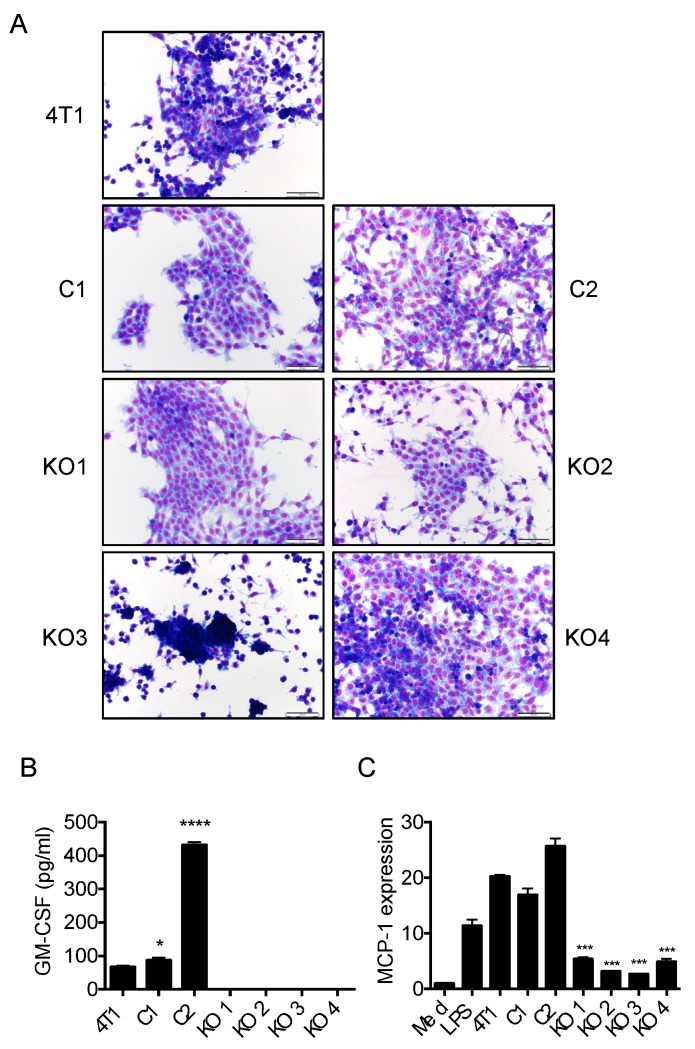
Characterization of GM-CSF-deficient 4T1 clones. (**A**) One thousand cells were seeded into a Lab-Tek II chamber slide (8 well) and cultured for 4 days and stained with Diff-Quik (Sysmex, Tokyo, Japan). The magnification of the original photos was 200×. The scale bar indicates 100 μm. (**B**) One hundred thousand 4T1 parental or clone cells in 2 mL RPMI1640 containing 10% FBS were seeded into 24-well plates and incubated at 37 °C for 3 days. The concentration of GM-CSF in each sample was measured by ELISA. The results are shown as the mean ± SD. *n* = 3 for each. * *p* < 0.05, **** *p* < 0.0001. (**C**) The induction of *Mcp-1* mRNA expression in mouse peritoneal macrophages by the culture supernatants of the parental 4T1 and clones was evaluated by qRT-PCR. LPS (10 ng/mL) was used as positive control. The results are shown as the mean ± SD. *n* = 3. *** *p* < 0.001.

**Figure 4 ijms-20-06342-f004:**
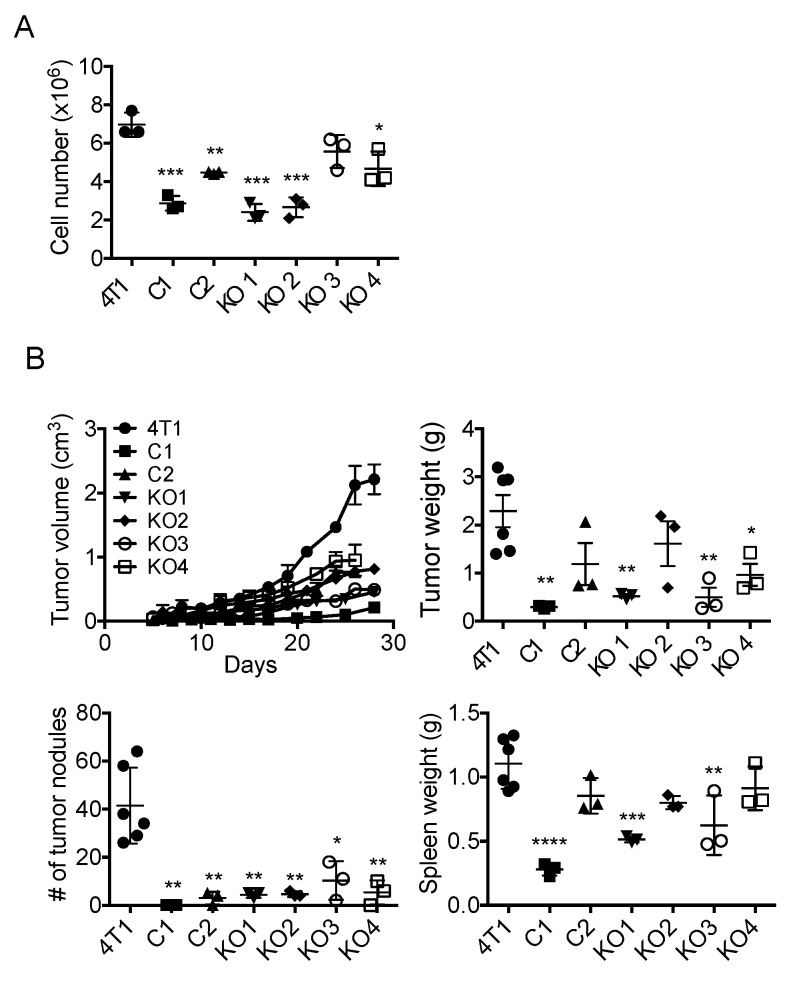
The growth of GM-CSF-deficient 4T1 clones in vitro and their progression in vivo. (**A**) One hundred thousand cells in 4 mL RPMI1640 containing 10% FBS were seeded into 6-well plates and incubated at 37 °C for 4 days and the number of cells was counted. The results are shown as the mean ± SD. *n* = 3. * *p* < 0.05, ** *p* < 0.01, *** *p* < 0.001. (**B**) One hundred thousand cells were injected into the mammary pad of mice and the sizes of tumors were measured and tumor volumes were calculated. Mice were euthanized 4 weeks after the injection and the weight of tumors, the number of lung metastases and the weight of spleens were evaluated. The results are shown as the mean ± SD. *n* = 3. * *p* < 0.05, ** *p* < 0.01, *** *p* < 0.001, **** *p* < 0.0001.

**Figure 5 ijms-20-06342-f005:**
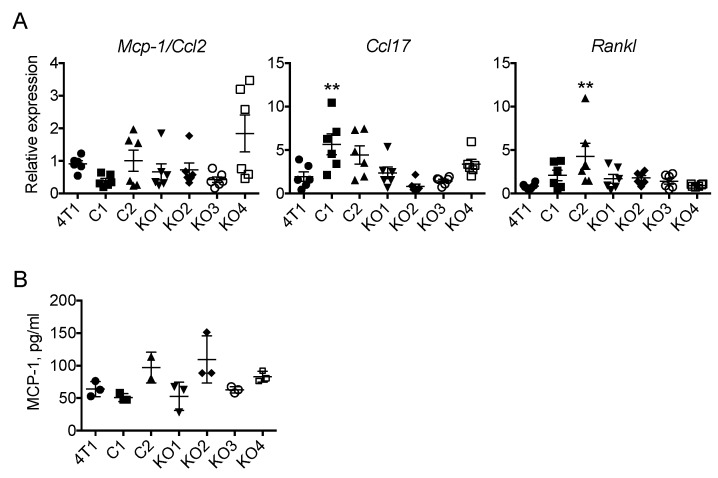
Expression of *Mcp-1*, *Ccl17* and *Rankl* mRNA in tumors of GM-CSF deficient 4T1 clones. One hundred thousand cells were injected into the mammary pad of mice. Mice were euthanized 2 weeks after the injection, tumors were harvested from the mice and total RNA was extracted. (**A**) The expression of each gene was examined by qRT-PCR. (**B**). The concentration of MCP-1 in sera was measured by ELISA. The results are shown as the mean ± SD. *n* = 3. ** *p* < 0.01.

**Figure 6 ijms-20-06342-f006:**
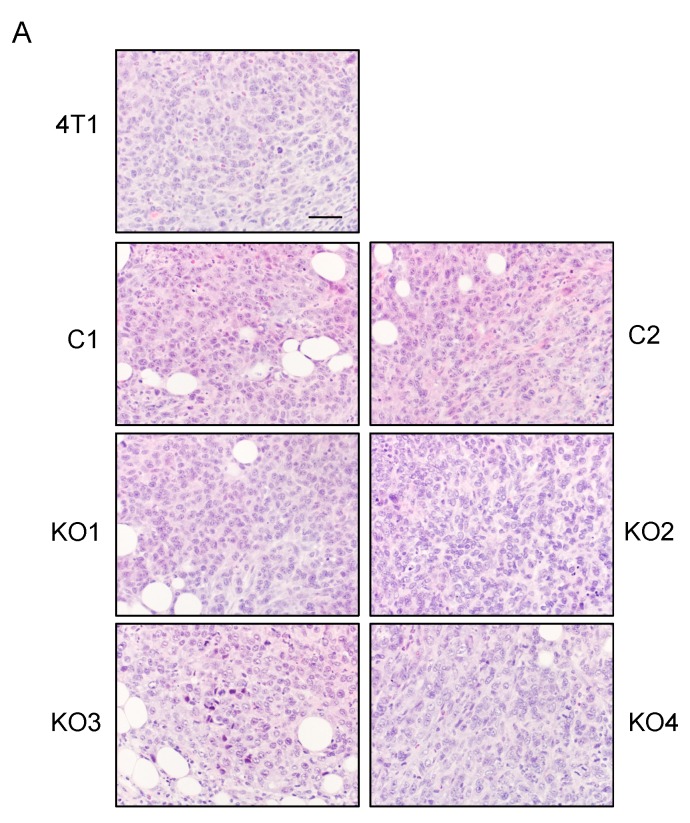
Infiltration of granulocytes, macrophages and Tregs in in tumors of GM-CSF-deficient 4T1 clones. One hundred thousand cells were injected into the mammary pad of mice, and tumors were harvested 2 weeks after the injection. Paraffin sections were prepared and subjected to H&E staining or immunohistochemistry. (**A**) H & E staining. The magnification of original photos was 200×. The scale bar indicates 100 μm. (**B**) The numbers of Ly6G-, F4/80- or FoxP3-positive cells in 5 randomly selected 400× fields were counted and the average number per high power field was obtained. The results are shown as the mean ± SD. *n* = 3.

**Figure 7 ijms-20-06342-f007:**
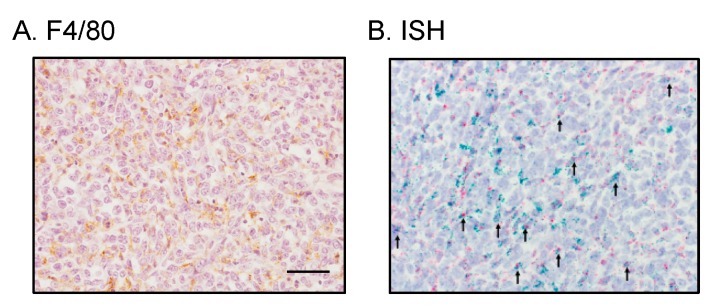
Detection of *Mcp-1* mRNA in macrophages infiltrating KO4 tumors. Tumors were harvested 2 weeks after the injection of KO4 cells and the expression of *Mcp-1* mRNA was examined by in situ hybridization. (**A**) F4/80 immunohistochemistry. (**B**) In situ hybridization. Green dots and red dots represent *Mcp-1* and *Adgre1 (F4/80)* mRNA, respectively. Arrows indicate cells with both *Mcp-1* and *Adgre1 (F4/80)* mRNA. The magnification of the original photos was 400×. The scale bar indicates 50 μm.

**Figure 8 ijms-20-06342-f008:**
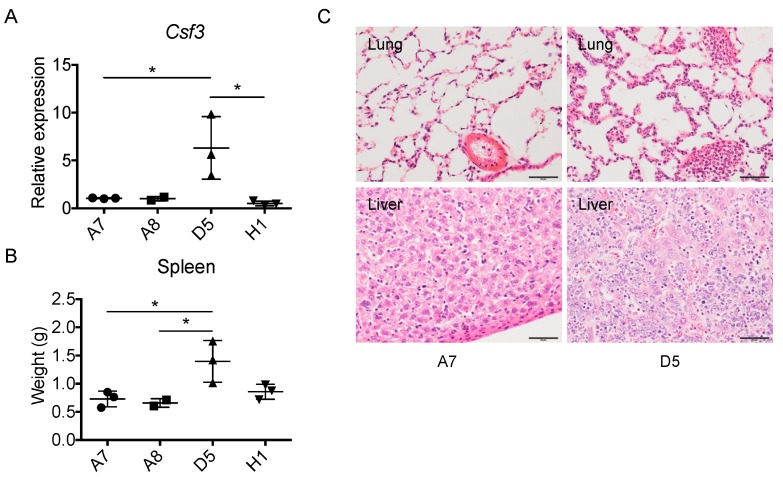
Correlation between the levels of *G-csf* (*Csf3*) mRNA expression by cancer cells and the degree of splenomegaly and tissue congestion. (**A**) Four 4T1 cell clones (1 × 10^5^ cells in 100 mL) were transplanted into the mammary pad of female BALB/c mice. Four weeks later, mice were euthanized, spleens were harvested from the mice and weights were measured. * *p* < 0.05. *n* = 3 for A7, D5 and H1, and *n* = 2 for A8. (**B**) The expression level of *Csf3 (Gcsf)* mRNA by each clone in vitro was evaluated by qRT-PCR. * *p* < 0.05. *n* = 3. (**C**) H&E staining of lung and liver tissue of mice bearing A7 or D5 clone cells. Capillaries and sinuses of lung and liver of D5 tumor-bearing mice were filled with leukocytes. The magnification of the original photos was 400×. The scale bar indicates 50 μm.

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
