# Peer review of "Cancer Cell-Derived Granulocyte-Macrophage Colony-Stimulating Factor Is Dispensable for the Progression of 4T1 Murine Breast Cancer"

_ijms, 2019, doi:10.3390/ijms20246342_

Round 1

Reviewer 1 Report

The manuscript "Cancer Cell-derived Granulocyte-macrophage 2 Colony-stimulating Factor is Dispensable for the 3 Progression of 4T1 Murine Breast Cancer" reports results of a study which is a follow up from a previous report on the GM-SCF secretion by 4T1 cells. The result is basically negative - the tumor-promoting effect of the tumor-derived GM-CSF predicted from the previous in vitro experiments was not confirmed. The study is methodologically sound and well-conducted. The evidence is sufficient, the results are unequivocal and support the conclusions. The observation that individual clones from the 4T1 cell line are not as good in establishing tumors as the parental cell line is interesting although it adds little information in view of the exquisite study by Wagenblast et al. 

The negative result is not a shortcoming - it is in line with the complex multifactorial mechanisms both of the establishment, progression and metastasizing of cancer and of the anti-tumor immunological mechanisms. It is in this aspect that I find an omission which reduces the enthusiasm for the paper. Like many other immunological factors, GM-CSF has an ambiguous role in tumor biology. Numerous studies have demonstrated its role in the maturation of the dendritic cells and for the induction of T cell responses to tumor antigens. I respect the humble and focused approach of the authors who apparently strife to stay very close to the data. Yet, not addressing experimentally and not even discussing the anti-tumoral function, the balance of the pro- and anti-tumoral activities, the role of mutation burden and tumor antigens in conjunction with the immunosuppressive environment makes the manuscript sound incomplete. 

In conclusion, this report has merits and should be published after some revision.

Author Response

Comments and Suggestions for Authors
The manuscript "Cancer Cell-derived Granulocyte-macrophage 2 Colony-stimulating Factor is Dispensable for the 3 Progression of 4T1 Murine Breast Cancer" reports results of a study which is a follow up from a previous report on the GM-SCF secretion by 4T1cells. The result is basically negative - the tumor-promoting effect of the tumor-derived GM-CSF predicted from the previous in vitro experiments was not confirmed. The study is methodologically sound and well-conducted. The evidence is sufficient, the results are unequivocal and support the conclusions. The observation that individual clones from the4T1 cell line are not as good in establishing tumors as the parental cell line is interesting although it adds little information in view of the exquisite study by Wagenblast et al. The negative result is not a shortcoming - it is in line with the complex multifactorial
mechanisms both of the establishment, progression and metastasizing of cancer and of the anti-tumor immunological mechanisms. It is in this aspect that I find an omission which reduces the enthusiasm for the paper. Like many other immunological factors, GM-CSF has an ambiguous role in tumor biology. Numerous studies have demonstrated its role in the maturation of the dendritic cells and for the induction of T cell responses to tumor antigens. I respect the humble and focused approach of the authors who apparently strife to stay very close to the data. Yet, not addressing experimentally and not even discussing the anti-tumoral function, the balance of the pro- and anti-tumoral activities, the role of mutation burden and tumor antigens in conjunction with the immunosuppressive environment makes the manuscript sound incomplete.
In conclusion, this report has merits and should be published after some revision.

(Our response)
We completely agree with the reviewer’s comment. We overlooked the important role of GMCSF in the development of some diseases and anti-tumor activities. In response to the reviewer’s comment, we have added new sentences in the Discussion section discussing the balance of pro and anti-tumor activities of GM-CSF. Thank you very much for pointing out our shortcoming.

Reviewer 2 Report

Yoshimura et al. tested the in vivo impact on GM-CSF knock out in the 4T1 breast model on its expression of CCL2, CCL17 and RANKL, based on their previous in vitro findings on macrophages.

While the research question is interesting, this papers lacks some fundamental experiments to enable significance of the results presented. Therefore I suggest to accept this manuscript after major revision of the following items:

Figure 1:

this figure is based on qRT-PCR results. However, underneath the qRT-PCR paragraph in the methods section, they only speak about RNA isolation from PEC. Does this mean that these data are derived from PEC or from whole tumor upon isolation (as described in legend of the figure)? If it was via PEC, the authors can not state they tested 'whole tumor'.  In case PEC was used, they should include data in each figure from healthy mice to see difference between PEC from a non-tumor bearing healthy mice vs the ones challenged with 4T1 or 4T1 GM-CSFKO clone It would be more interesting to sort the stromal cells from the tumor cells and evaluate mRNA expression AND protein expression of GM-CSF, CCL2, CCL17 and RANKL for each sorted population. Especially as the authors state that they previously showed clear decrease in macrophage mRNA expression levels upon anti-GMCSF in vitro. So the lack of differences they observe, could be the result of a decrease in macrophages concomitant with an increase in the 4T1 cells or the other way around... Or in other words, the whole-tumor based analysis of mRNA can hide variabilities in mRNA expression levels between different subsets within the tumor It was previously shown that anti-GM-CSF administration (GSK3196165) in patients with rheumatoid arthritis, did result in CCL17 reduction in the serum (https://acrabstracts.org/abstract/a-phase-iia-mechanistic-study-of-anti-gm-csf-gsk3196165-with-methotrexate-treatment-in-patients-with-rheumatoid-arthritis-ra-and-an-inadequate-response-to-methotrexate/). Therefore authors should verify if anti-GM-CSF treatment does not show any effects on the serum of CCL2, RANKL and CCL17.

Figure 5:

Amount of GM-CSF in serum (at protein level) of mice with GM-CSF ko 4T1 cells should be measured and compared with that of mice challenged with wild type 4T1 cells to find out if no other cells take over GM-CSF production of the 4T1 cells.

Figure 7:

Evaluation of different markers on protein level upon single cell suspension with flow cytometry will be the best strategy to analyse overall % and mean fluorescence intensities linked to markers like F4/80, CCL2,...

Figure 8:

While GM-CSF expression was measured on protein level in Figure 3, this was not the case for G-CSF, so not 100% sure conclusions can be drawn based on mRNA expression levels. Only via a knock out of G-CSF within D5 the authors can claim that G-CSF specifically causes splenomegaly. With the experiments performed here, is can not be ruled out that other proteins are maybe up-or downregulated within D5 in comparison with the other clones, causing the splenomegaly.

Author Response

Comments and Suggestions for Authors
Yoshimura et al. tested the in vivo impact on GM-CSF knock out in the 4T1 breast model on its expression of CCL2, CCL17 and RANKL, based on their previous in vitro findings on macrophages. While the research question is interesting, this papers lacks some fundamental experimentsto enable significance of the results presented. Therefore I suggest to accept this manuscript after major revision of the following items:

Figure 1:
this figure is based on qRT-PCR results. However, underneath the qRT-PCR paragraph in the methods section, they only speak about RNA isolation from PEC. Does this mean that these data are derived from PEC or from whole tumor upon isolation (as described in legend of the figure)? If it was via PEC, the authors can not state they tested 'whole tumor'. In case PEC was used, they should include data in each figure from healthy mice to see difference between PEC from a non-tumor bearing healthy mice vs the ones challenged with 4T1 or 4T1 GM-CSFKO clone It would be more interesting to sort the stromal cells from the tumor cells and evaluate mRNA expression AND protein expression of GM-CSF, CCL2, CCL17 and RANKL for each sorted population. Especially as the authors state that they previously showed clear decrease in macrophage mRNA expression levels upon anti-GMCSF in vitro. So the lack of differences they observe, could be the result of a decrease in macrophages concomitant with an increase in the 4T1 cells or the
other way around... Or in other words, the whole-tumor based analysis of mRNA can hide variabilities in mRNA expression levels between different subsets within the tumor It was previously shown that anti-GM-CSF administration (GSK3196165) in patients with rheumatoid arthritis, did result in CCL17 reduction in the serum
(https://acrabstracts.org/abstract/a-phase-iia-mechanistic-study-of-anti-gm-csfgsk3196165-with-methotrexate-treatment-in-patients-with-rheumatoid-arthritis-ra-and-aninadequate-response-to-methotrexate/). Therefore authors should verify if anti-GM-CSF treatment does not show any effects on the serum of CCL2, RANKL and CCL17.

(Our response)
First, we thank the reviewer for spending invaluable time to review our manuscript and providing helpful comments. We apologize for not providing sufficient information for the better understanding of our results. RNA from PEC was used only in a study presented as Figure 3C. In the paragraph under “Transplantation of 4T1 cells and neutralization of GM-CSF”, we stated that tumors were harvested and fixed in RNAlater, but failed to state that they were used to isolate RNA for RT-qPCR. We have added the information that RNA was also isolated from tumor tissues in the M&M section. We simply made a mistake.
Regarding the serum concentration of CCl2, RANKL and CCL17, we previously reported that the serum level of MCP-1/CCL2 did not change after treatment with anti-GM-CSF antibody (Yoshimura et al., Ref 7 in this manuscript). Since serum CCL2 concentration did not change after anti-GM-CSF treatment, in the present study we focused on the local expression of CCL2 and others in the 4T1 tumor microenvironment. From these reasons, we did not measure the serum level of CCL17 or RANKL. However, the reviewer’s comment is very interesting and we
would like to address the question in our future study.

Figure 5:
Amount of GM-CSF in serum (at protein level) of mice with GM-CSF ko 4T1 cells should be measured and compared with that of mice challenged with wild type 4T1 cells to find out if no other cells take over GM-CSF production of the 4T1 cells.

(Our response)
We appreciate the reviewer’s comment. As pointed out by the reviewer, it is quite possible that other cell types produce GM-CSF. We measured serum concentration of GM-CSF in mice inoculated with parental (WT), control or GM-CSF KO clones by ELISA. There was no detectable level of GM-CSF in the sera from all mice. Petráčková et al. (Int J Oncology. 2012;40:1915-1922; Ref. 20 in this manuscript) previously transduced a leukemic cell line to overexpress GM-CSF and inoculated them into mice. GM-CSF was not detectable in the sera from normal mice or mice inoculated with the parental cells but detectable in mice inoculated with GM-CSF-transduced cells. Their transduced cells produced 100 ng/106 cells/24 hours, which was probably 100-fold higher than 4T1 cells did. Thus, our result is consistent with that by Petráčková et al., and it is unlikely that other cell types take over 4T1 cells and produce significant amounts of GM-CSF. Since our focus was to examine the direct effect of cancer cellderived GM-CSF to the 4T1 tumor microenvironment, we did not include this information in the original manuscript. In the revised manuscript, we added this information (Line 180-182). Thank
you very much for the important comment.

Figure 7:
Evaluation of different markers on protein level upon single cell suspension with flow
cytometry will be the best strategy to analyse overall % and mean fluorescence intensitieslinked to markers like F4/80, CCL2,...

(Our response)
We appreciate the reviewer’s comment and suggestion to use flowcytometry. We initially used immunohistochemistry (IHC) to detect MCP-1-prodicing cells in 4T1 tumors. However, our attempts with multiple primary antibodies against mouse MCP-1 were not successful. MCP-1 is produced at a high level by a wide variety of cells, including macrophages, but it is secreted from the cells very efficiently, making identification of MCP-1 inside the cells by IHC difficult. On the other hand, MCP-1 mRNA is abundant in MCP-1-producing cells. Therefore, we decided to
use in situ hybridization (ISH). ISH is an established method to determine cell types expressing a certain gene in a tissue. Flowcytometry is a powerful method and would provide more quantitative result if the method is established. Considering the efficient secretion of MCP-1 by macrophages, additional in vitro culture in the presence of an inhibitor that inhibits MCP-1 secretion, such as brefeldin A, may be required. This would not be a good method for our purpose because macrophages that were negative could become positive during the culture. In this study we only wanted to show that MCP-1 could be produced by tumor-infiltrating
macrophages independent of cancer cell-derived GM-CSF; therefore, we chose to use ISH. This information is now added to the text. It would be great if we can detect MCP-1-producing cells by flowcytometry in future studies. We thank the reviewer for the interesting suggestion.

Figure 8:
While GM-CSF expression was measured on protein level in Figure 3, this was not the case for G-CSF, so not 100% sure conclusions can be drawn based on mRNA expression levels. Only via a knock out of G-CSF within D5 the authors can claim that G-CSF specifically causes splenomegaly. With the experiments performed here, is can not be ruled out that other proteins are maybe up-or downregulated within D5 in comparison with the other clones, causing the splenomegaly.

(Our response)
We understand the concern raised by the reviewer. Since the deletion of GM-CSF in cancer cells did not affect splenomegaly, we explored the possibility that cancer cell-derived G-CSF may contribute to splenomegaly and tissue congestion. We found a correlation between the level of G-CSF expression and splenomegaly and tissue congestion, and suggested a role of G-CSF in splenomegaly and tissue congestion in this model. As introduced in the Discussion section, G-CSF was recently knocked out in 4T1 cells by others (Ravindranathan et al., Ref.22 in the
revised manuscript) and G-CSF-deficiency markedly reduced splenomegaly in 4T1 tumor-bearing mice, indicating that G-CSF produced by 4T1 cells significantly contributes to the development of splenomegaly. Therefore, we did not perform the same experiment to further support our hypothesis that 4T1 cell-derived G-CSF plays a role in splenomegaly. In response to
the reviewer’s comment, we revised our discussion about the role of G-CSF in splenomegaly and tissue congestion in this model. Thank you very much for the comment.

Round 2

Reviewer 2 Report

As all comments were addressed correctly, I suggest to accept the revised manuscript in its present form.